# Promising High-Tech Devices in Dry Eye Disease Diagnosis

**DOI:** 10.3390/life13071425

**Published:** 2023-06-21

**Authors:** Andrea De Luca, Alessandro Ferraro, Chiara De Gregorio, Mariateresa Laborante, Marco Coassin, Roberto Sgrulletta, Antonio Di Zazzo

**Affiliations:** Ophthalmology Complex Operative Unit, University Campus Bio-Medico, 00128 Rome, Italy

**Keywords:** dry eye disease, diagnostic device, ocular surface

## Abstract

Background: Dry eye disease (DED) is a common and debilitating condition that affects millions of people worldwide. Despite its prevalence, the diagnosis and management of DED can be challenging, as the condition is multifactorial and symptoms can be nonspecific. In recent years, there have been significant advancements in diagnostic technology for DED, including the development of several new devices. Methods: A literature review of articles on the dry eye syndrome and innovative diagnostic devices was carried out to provide an overview of some of the current high-tech diagnostic tools for DED, specifically focusing on the TearLab Osmolarity System, DEvice Hygrometer, IDRA, Tearcheck, Keratograph 5M, Cornea Dome Lens Imaging System, I-PEN Osmolarity System, LipiView II interferometer, LacryDiag Ocular Surface Analyzer, Tearscope-Plus, and Cobra HD Camera. Conclusions: Despite the fact that consistent use of these tools in clinical settings could facilitate diagnosis, no diagnostic device can replace the TFOS algorithm.

## 1. Introduction

The TFOS DEWS II (Tear Film and Ocular Surface Society International Dry Eye Workshop II, 2017) defines dry eye disease (DED) as “a multifactorial disease of the ocular surface characterized by a loss of homeostasis of the tear film and accompanied by ocular symptoms, in which tear film instability and hyperosmolarity, ocular surface inflammation and damage, and neurosensory abnormalities play etiological roles” [1]. The prevalence of DED exhibits a positive correlation with advancing age and varies between five percent and fifty percent across the overall population [2]. DED is characterized by a range of symptoms such as ocular pain, burning, stinging, discomfort, a foreign body sensation, poor visual acuity, photophobia, and irritation [1,2]. The symptoms of dry eye disease can range from minor discomfort to substantial grievances that interfere with daily functioning, decrease quality of life, and carry notable consequences for the socioeconomic structure [1,2].

The first phase in the process of diagnosing dry eye disease involves the utilization of triage questions, which could establish the need for additional DED evaluation and exclude disorders such as conjunctivitis, blepharitis, Sjögren syndrome, infection, and lid-related disease.

Based on the TFOS DEWS II, a dry eye diagnosis requires the patient to score positively on one of two specific symptom questionnaires (DEQ-5, Dry Eye Questionnaire-5 score ≥ 6 or OSDI, Ocular Surface Disease Index score ≥ 13). This must be followed by the presence of a minimum of one positive clinical sign, such as decreased tear film stability (NIBUT, non-invasive tear break-up time, <10 s), elevated tear osmolarity (>308 mOsm/L), significant inter-eye disparity in osmolarity (>8 mOsm/L), or ocular surface damage indicated by fluorescein and lissamine green (>5 corneal spots, >9 conjunctival spots, or lid margin ≥ 2 mm length and ≥25% width).

OSDI refers to a verified questionnaire that is commonly employed in clinical trials due to its ability to provide a rapid assessment of dry eye disease (DED) and its impact on the quality of life (QoL) of patients [3]. The OSDI consists of 12 items that aim to evaluate the symptoms experienced by patients during the preceding week. It is organized into three sections: the occurrence of symptoms, the impact on vision-related quality of life, and the identification of any environmental triggers [3]. Each item is evaluated using a scale ranging from zero to four. The scale used to measure frequency is as follows: none of the time (0), some of the time (1), half of the time (2), the majority of the time (3), and the entire time (4). The overall numerical value is calculated utilizing a range that extends from zero to one hundred, whereby elevated scores denote heightened levels of impairment. A score of 0 to 12 is regarded as normal; a score of 13 to 22 suggests mild disease; a score of 23 to 32 denotes moderate DED; and a score of 33 to 100 indicates severe DED [3]. Furthermore, OSDI does not discriminate between evaporative and aqueous deficiencies [4].

It is possible that the TFOS DEWS II diagnostic algorithm is not the most effective method when used in a clinical environment, despite the fact that it provides both a comprehensive and full approach to detecting dry eye disease. Although it offers a complete evaluation, the diagnostic procedure typically consists of a number of steps that can be challenging to carry out in fast-paced clinical settings with limited time.

Therefore, in this paper, we examine new high-tech imaging systems for ocular surface evaluation. These systems claim to have several benefits over traditional methods of diagnosis, such as being non-invasive, providing standardized and objective results, enabling the monitoring of disease progression and treatment effectiveness, being user-friendly, and enabling rapid task execution. However, regardless of the studies reviewed, the reliability of these devices is low.

## 2. Materials and Methods

A literature review of articles on dry eye syndrome and innovative diagnostic devices published in the last 15 years and available on the National Library of Medicine was carried out without any restriction of language, especially focusing on the TearLab Osmolarity System, IDRA, Tearcheck, Keratograph 5M, I-Pen Osmolarity System, Lipiview II Interferometer, LacryDiag Ocular Surface Analyzer, Cornea Dome Lens Imaging System, DEvice Hygrometer, Tearscope-Plus, and Cobra HD Camera. All published peer-reviewed randomized clinical trials, meta-analyses, systematic reviews, and observational studies about these topics were evaluated. Table 1 shows all the devices analyzed and the exams they perform.

TearLab Osmolarity System^®^ (TearLab Corporation, San Diego, CA, USA) is a non-invasive diagnostic device that analyzes the osmolarity of a patient’s tears. Osmolarity refers to the total concentration of dissolved substances present in a solution without regard to their density, size, molecular weight, or electrical charges. The process of tears evaporating, a decrease in the production of tears, and the dysfunction of the meibomian gland are all factors that contribute to an elevation in tear osmolarity. The evaluation of tear osmolarity is considered a highly effective diagnostic tool for every type of dry eye syndrome [5]. The tear film of the exposed ocular surface area exhibits a lower osmolarity. Various environmental factors such as wind, cigarette smoke, indoor air conditioning, and heat, as well as prolonged computer use leading to reduced blinking frequency, have been identified as potential impediments to evaporation, thereby affecting tear osmolarity [6]. Tear osmolarity was also found to correlate with increased concentrations of inflammatory cytokines and matrix metalloproteinases (MMPs), as well as HLA-DR (Human Leukocyte Antigen-DR) overexpression, suggesting that tear osmolarity could potentially serve as a predictive measure for ocular surface diseases that are linked to high levels of inflammatory mediators [6].

The TearLab Osmolarity System is composed of a set of instruments, including a reader device, pens, and test cards. The reader is a small countertop device that calculates and shows the osmolarity test results on a liquid crystal display. The TearLab osmolarity device is equipped with a pair of pens that are used to hold the test card and transmit the data to the reader. The test card is attached to the pens and takes a sample of 50 nL in milliosmoles per liter (mOsm/L) units. The contact between the tear film and the eyelid occurs at the temporal margin. After hearing the beep confirming successful tear collection, the pen is docked into the reader [7].

This osmometer has several advantages, including being a small, portable device that can be used in a doctor’s office and requiring less than 100 nL of tear fluid [5]. Furthermore, it uses a temperature-corrected tear fluid impedance measurement, enabling an indirect evaluation of tear osmolarity and providing precise results within a brief timeframe [5,8]. It may also be used in combination with other diagnostic tools, such as the LipiView system, to provide a more detailed image of a patient’s dry eye condition. According to Szczesna-Iskander, it is necessary to take a minimum of three consecutive measurements to obtain clinically trustworthy tear osmolarity values using the TearLab Osmolarity System. The utilization of the highest osmolarity value to identify DED requires careful consideration due to the frequent occurrence of anomalous readings of tear osmolarity [9]. Nevertheless, Szalai et al. found significant overlap in tear osmolarity values measured with the TearLab system in the control (22 healthy individuals) and dry eye groups (21 patients with non-Sjögren syndrome dry eye (NSSDE) and 20 patients with Sjögren syndrome dry eye (SSDE)), implying that measurement of tear osmolarity utilizing the TearLab osmometer is highly variable and does not distinguish individuals diagnosed with dry eye disease from healthy controls (mean tear osmolarity was 296.77 ± 16.48 mOsm/L in NSSDE, 303.36 ± 17.22 mOsm/L in SSDE, and 303.52 ± 12.92 mOsm/L in the control group; *p* = 0.018) [10].

As a result, TearLab is a quick tool in clinical practice, but its usefulness is limited by the literature-reported low reliability in recognizing DED and its primary use in evaporative dry eye.

IDRA^®^ Ocular Surface Analyzer (SBM SISTEMI, Inc., Torino, Italy) is a diagnostic tool that uses infrared and ultraviolet light to evaluate the health of the ocular surface. Changes in the tear film and meibomian glands, which can be indicative of DED, can be detected by the instrument. The instrument is able to identify MG as well as all three layers of the tear film (lipid, aqueous, and mucin). This allows physicians to determine which parts of the tear film require treatment based on the type of insufficiency. IDRA must be placed between a slit lamp and a biomicroscope. Its pins have been designed to fit perfectly into the hole left by removing the plate used for the tonometer, and it conducts a 5-minute non-invasive test [11]. The instrument produces a beam of white light onto the corneal surface, and the resultant reflection of light from the tear film presents a white, fan-shaped region that covers the inferior third of the cornea [12]. The five parameters include NIBUT, TMH, lipid layer interferometry, ocular blink quality, and infrared meibography. The non-invasive break-up time (NIBUT) is determined through the utilization of Placido’s disc to project ring patterns on the cornea, followed by the measurement of the duration in seconds between the complete blink and the first disturbance of the reflected image on the cornea [11]. The utilization of infrared illumination in non-invasive meibography has the capability to identify morphological alterations in the meibomian gland. On the other hand, tear interferometry can be employed to assess the lipid layer of the tear film. The evaluation of meibomian gland morphology offers valuable clinical evidence for the diagnosis of evaporative DED, while assessments of the lipid layer of the tear film facilitate the monitoring of meibomian gland function [10]. The photograph shows the identification of the upper and lower tear meniscus as well as the evaluation of tear meniscus height along the lower lid margin [11]. A prospective study with 75 patients (40 with DED and 35 healthy subjects) found good concordance between the IDRA ocular surface analyzer and standard diagnostic procedures in differentiating between individuals with normal ocular function and those with dry eye disease. It had an area under the curve of 0.868 (95% confidence interval: 0.809–0.927) to detect DED [13].

The lipid layer is important for regulating the evaporation of the tear film. A test for lipid layer pattern (LLP) evaluation is based on interference phenomena, but it is influenced by subjective interpretation of the patterns [14]. The lipid layer thicknesses are determined through the utilization of Dr. Guillon’s international grading system, which enables the calculation of the average, maximum, and minimum thicknesses of the lipid layer pattern grades [15]. The grades are converted to nanometer units and classified into a range of 15 to 100 nm according to the observed patterns. The maximum cutoff wavelength of IDRA is 100 nm [16].

Despite the fact that IDRA has the benefit of assessing multiple ocular surface parameters with a single device, the findings are contradictory. In a retrospective cross-sectional study with 47 non-Sjögren dry eye patients, Jeon et al. demonstrated a significant correlation between dry eye symptoms and the partial blink rate as well as meibomian gland dropout rates measured using IDRA. On the other hand, Lee et al. demonstrated that IDRA exhibited a considerably lower percentage of meibomian gland dropout and a greater partial blink rate than another device, LipiView^®^ II, in a cross-sectional, single-visit observational study with 47 participants [11,12]. These findings suggest that these devices should not be used interchangeably when assessing meibomian gland dropouts and partial blink rates [11]. Rinert et al. demonstrated a favorable correlation between everyday clinical diagnostic examinations. The researchers found that the utilization of pathologic meibography, interferometry, and tear meniscus measurements with the analyzer produced a 96% estimated probability of dry eye disease. Simultaneously, the percentage of eyes exhibiting pathological observations in the three sets of examinations was relatively minimal, suggesting that dry eye disease (DED) may manifest in diverse clinical presentations and necessitate a comprehensive assessment [13].

Thus, IDRA appears to be an interesting device for diagnosing DED because it evaluates all three components of the tear film; however, the literature presents conflicting and limited findings.

Tearcheck^®^ (E-Swin, Houdan, France) is a stand-alone device with an integrated screen that allows the user to view all acquisitions and exams in real time. The device facilitates quick evaluations that include nine examinations: non-invasive break time, tear film stability, ocular surface inflammatory assessment, meibography IR, Demodex, eye redness, abortive blinking, tear meniscus height, and the OSDI questionnaire. This results in a simple dry eye analysis.

Using the Demodex exam, an enlarged image of the base of the eyelashes can be obtained, allowing for the tracing and visualization of signs indicating the presence of Demodex mites. As a result, the device takes high-resolution images of the ocular surface, enabling the detection of changes in the cornea, conjunctiva, and tear film, such as the existence of inflammation or ocular surface damage (dry spots or erosion).

Although the device is non-invasive and simple to use, there is no evidence in the literature to support its use in the diagnosis and management of DED.

Keratograph 5M^®^ (Oculus, Wetzlar, Germany) is a diagnostic device that uses non-invasive imaging technology to evaluate the health of the cornea. The device can identify changes in the surface and shape of the cornea, which can be indicative of DED. It is a corneal topographer that is equipped with a real keratometer and a color camera. Its purpose is to capture external images by projecting a ring pattern from a placido disc on the tear film surface using an infrared light source. It may evaluate non-invasive break-up time (NIBUT), meibography, bulbar redness, tear meniscus height, lipid layer (interferometry), and tear film dynamics (monitoring of tear film particle flow, from which inferences about tear film viscosity can be inferred).

The findings of this tool in the literature are also contradictory. On the one hand, the keratograph has significant examiner bias [17], but on the other hand, it has been reported to have strong inter-examiner reproducibility (mean difference between examiners of 0.08 ± 0.55 and 0.13 ± 0.50 grade units in two separate sessions, respectively) with low within-subject variability (95% limits of agreement for two different examiners of −1.02 to +1.10 and −1.27 to +1.09 grade units, respectively) [18]. In a prospective study with 42 patients with DED and 42 healthy subjects, Tian et al. found that utilizing the non-invasive Keratograph tear break-up time (NIKBUT) and tear meniscus height (TMH) measurements through the K5M device could serve as a straightforward and non-invasive method for screening dry eye while also exhibiting satisfactory levels of repeatability and reproducibility (coefficient of variation (CV%) ≤ 26.1% and intraclass correlation coefficient (ICC) ≥ 0.75 for all measurements). Nevertheless, it was observed that NIKBUTs exhibited greater reliability in individuals with dry eye disease (DED) as compared to TMH [19]. 

Sutphin et al. concluded that keratographic measures cannot be considered interchangeable alternatives for commonly used clinical measures. Furthermore, they found that there is no specific test that can provide objective support for the diagnosis [20]. Indeed, according to Pérez-Bartolomé et al., the Keratograph 5M was observed to overestimate ocular redness scores in comparison to subjective grading scales when utilized for the purpose of assessing the degree of ocular redness [21]. Furthermore, in an observational cross-sectional study of 47 subjects with DED and 41 normal control subjects, Chen et al. showed a limited association between the keratograph tear meniscus height (TMH) and Schirmer scores among individuals with dry eye disease (DED). They also demonstrated that in the comparison of Fourier-domain optical coherence tomography (FD-OCT) and the Keratograph 5M, both instruments exhibited notable diagnostic precision in distinguishing between normal patients and those with dry eye disease. However, it was observed that the FD-OCT measurements of tear meniscus height (TMH) were more dependable than the keratograph data in the DED group [22]. Specifically, while the keratograph and FD-OCT measurements of TMH were closely correlated, the former tended to yield lower measurements than the latter [23].

Despite the fact that the Keratograph 5M is a non-invasive diagnostic topographer for DED, it is not yet a substitute for multiple clinical tests such as the Schirmer test and FBUT because its reliability is very weak.

I-PEN Osmolarity System^®^ (I-MED Pharma Inc., Dollard-des-Ormeaux, QC, Canada) is a portable electrical tool that assesses the osmolarity of tears by measuring the electrical impedance of the eye tissues on the palpebral conjunctival membrane. The occurrence of an inflammatory cascade at the ocular surface is initiated by tear film hyperosmolarity, which ultimately leads to the loss of goblet cells, epithelial injury, and the production of cytokines. This condition is responsible for causing ocular distress and vision impairment in patients with DED. The I-PEN’s usefulness has been questioned in the literature, though numerous studies have considered the I-Pen appropriate and reliable for clinical use [24,25,26]. Park et al. concluded that the I-Pen osmometer demonstrates favorable performance in diagnosing DED in clinical settings; however, it should not be solely relied upon for evaluating DED. Nonetheless, the I-Pen osmometer can serve as a valuable tool for screening and identifying dry eye disease [24]. In contrast, some researchers failed to establish any correlations between tear film osmolarity values acquired through the I-PEN system and various subjective or objective parameters of dry eye disease (DED). Furthermore, they showed that the I-PEN system was less effective than the TearLab Osmolarity System in delineating subjects with and without dry eye disease [7,27,28]. Shimazaki et al. found no statistically significant difference in mean tear film osmolarity between the DED (871 eyes) and non-DED (51 eyes) groups using the I-PEN system (294.76 ± 16.39 vs. 297.76 ± 16.72 mOsms/L, respectively, *p* = 0.32). Furthermore, motion may affect osmolarity readings acquired through the I-Pen system; however, the influence of this factor can be minimized if the measurements are carried out by a highly trained clinician [29]. Alanazi et al. evaluated the relationship between osmolarity results acquired by the TearLabTM and I-Pen^®^ systems in individuals with a high body mass index (BMI). The I-Pen^®^ results (294–336 mOsm/L in the study group of 30 male subjects with a high BMI and 278–317 mOsm/L in the control group of 30 healthy males) were significantly higher than the TearLab^TM^ scores (278–309 mOsm/L in the study group and 263–304 mOsm/L in the control group). Furthermore, the outcomes obtained from the I-Pen^®^ measurements exhibited significant variations in osmolarity values and demonstrated a considerable lack of accuracy in distinguishing normal eyes as compared to the TearLab^TM^ system [29].

As a result, the I-PEN is a portable, easy-to-use, autocalibrated device that is not affected by variations in tear film volume and requires only a simple touch of the palpebral conjunctiva. Nevertheless, it can only detect tear osmolarity, and further investigations are required to determine its utility.

LipiView^®^ II interferometer (TearScience Inc., Morrisville, NC, USA) is a device for ocular imaging that examines the interferometric pattern of the tear film. It achieves this by measuring the lipid layer thickness (LLT) of the tear film with nanometer accuracy; however, it has an upper cut-off to assess LLT values of 100 nm [11]. In addition to that, it records the dynamics of blinking and images the structure of the meibomian gland. Compared to IDRA, in a cross-sectional single-visit observational study with 47 participants, Min Lee et al. found no significant difference in LLT. However, IDRA had a considerably lower rate of MG dropout and a higher PBR (IDRA MG dropout 45.36 ± 21.87 and PBR 0.23 ± 0.27; LipiView^®^ II interferometer MG dropout 36.51 ± 17.53 and PBR 0.51 ± 0.37) [11]. In comparison to Keratograph 5M (K5M), Wong et al. showed that LVII exhibited a statistically significant reduction in meiboscores and a lower percentage of MG dropout in 20 subjects (1.43 ± 0.78 vs. 1.90 ± 0.81, *p* = 0.001) [30]. These results suggest that there is poor interchangeability between the methods used to evaluate DED features, particularly MG dropouts and PBR.

In conclusion, the data obtained from the LVII LLT should be compared to other instruments. However, additional studies with larger sample sizes are necessary.

LacryDiag Ocular Surface Analyzer (Quantel Medical, Cournon-d’Auvergne, France) is an ophthalmic device used to diagnose and monitor the tear film and meibomian glands. It takes non-invasive photographs with white or infrared light to assess the height of the lower tear meniscus, the distance between the upper and lower eyelids, tear film interferometry, and non-invasive tear film break-up time. Toth et al. demonstrated that it is a non-invasive, simple-to-use device able to analyze the tear film and save photos for later use [31]. Despite this result, there is a great deal of variability between measurements performed by this instrument and those performed by another innovative device, such as the OCULUS Keratograph 5M, possibly reflecting differences in image processing or the need for subjective evaluation by the observer for a considerable number of these measurements [32,33]. Ward et al., for instance, evaluated the repeatability of the LacryDiag Ocular Surface Analyzer for both intra- and inter-observer measurements and compared it to the OCULUS Keratograph 5M in 30 healthy subjects. Their findings revealed a good relationship between the devices (no differences in mean values for tear meniscus height, NIBUT, or tear film interferometry, except for lipid layer interferometry), but low agreement for any individual observer (intra-observer variability for NIBUT was significantly higher for the Keratograph, *p* = 0.0003 for observer A and *p* < 0.0001 for observer B). According to the authors, the observed inconsistency could be attributed to the utilization of repeated testing and the inclusion of subjects without dry eye conditions. Therefore, the authors concluded that in the identification and follow-up of patients with dry eye disease (DED), it is essential to consider the reproducibility of the testing instrument and the utilization of different outcome measures [33].

In summary, LacryDiag is a promising instrument for assessing the ocular surface, but there is a lack of research about it in the medical literature.

Cornea Dome Lens Imaging System^®^ (Occyo, Innsbruck, Austria) is an imaging system that attempts to provide uniform ocular surface color photographs respecting position, illumination, focus, and operator independence. This is achieved by overcoming the limitations of objective methods that are based on digital ocular surface images. The device is composed of a novel imaging lens that conforms accurately to the curvature of the eye, enabling high-resolution imaging of the visible ocular surface. In addition, the device incorporates a fixation target that guarantees a centralized view into the lens, thereby minimizing eye movements. Furthermore, the set is composed of software designed for eye tracking as well as an illumination unit. To maintain the stability of the patient’s head, a chin rest and a forehead band are utilized. The aim of the system is to obtain photographs in a standardized manner without the need for human intervention. Therefore, according to Lins, this tool possesses the ability to evaluate the extent of bulbar eye redness in an objective and replicable manner, utilizing the images it has captured [34].

This technology has the potential to provide an objective technique based on the digital ocular surface for assessing bulbar eye redness, overcoming the limitations of subjective photographic scales that suffer from inter-rater variability. However, its role in the DED diagnostic process should be investigated in future studies.

DEvice Hygrometer© (AI, Rome, Italy) is a complete, low-cost diagnostic-therapeutic tool for ocular surface management that has the capability to quickly identify the entities of production, clearance, and stability, along with the severity of tear film evaporation, and drives the subsequent therapy through the utilization of simple algorithms. The device operates by detecting changes in the relative humidity (RH) levels within a confined environment surrounding the ocular surface (Figure 1).

The diagnostic component of the device works by measuring the evaporation of the tear film from the ocular surface at a variable speed that may be modified by the user. At a certain temperature, the device measures the baseline and post-stimulus relative humidity values. The sensor is placed in a cup on the orbital edges by the operator. The measurements are carried out in a closed environment made up of the cup and the ocular surface system. Using the acquired data, it is possible to construct progression curves for relative humidity (RH) that have been corrected for temperature. The collected data are “basal” values that are combined with measurements taken in reaction to diverse kinds of stimuli, such as air blows, alterations in temperature within the microenvironment surrounding the ocular surface, light stimuli, and so on. Additionally, a non-contact sample mechanism built into the device allows for the collection of a certain amount of tear evaporation. Despite the fact that incomplete blinking and tear clearance may have an impact on the accuracy of measurements obtained from the DEvice^©^, it represents a low-cost, efficient, accurate, rapid, and safer (as it is non-contact) instrument for measurement of tear films. Indeed, a preliminary observational pilot study with 8 patients (2 with DED and 6 healthy subjects) has shown that individuals with dry eye disease (DED) showed higher relative humidity values compared to healthy individuals. However, additional studies involving a larger sample size are necessary to confirm these findings. The diagnostic device exhibits potential for local drug nebulization, thereby presenting an option for alternative therapeutic applications in the future [35].

Thus, even though it is a promising diagnostic tool, its use in clinical practice is now limited to evaporative dry eye.

Tearscope-Plus^TM^ (Keeler, Windsor, UK) is a relatively new portable device that may be connected to a slit lamp for conducting non-invasive assessments of the tear film. It allows the examination of the interference patterns of the lipid layer across the whole cornea without the need for fluorescein, thereby enabling the evaluation of non-invasive break-up time (NIBUT), tear meniscal height (TMH), and lipid layer thickness of the tear film (LLT). Guillon’s classification is employed for the purpose of determining the thickness of the lipid layer (LLP, lipid layer pattern). The system includes five different types of lipid layer patterns: open meshwork (OM), closed meshwork (CM), wave (W), amorphous (AM), and color fringe (CO). In addition to normal LLPs and events, atypical ones were described. This method is effective for investigating the quality and structure of the tear film; nevertheless, it is dependent on the observer’s judgment, which can be affected by the sort of pattern that is viewed [36]. Visualizing thicker lipid layers can be challenging due to the lack of distinguishing morphological traits and color fringes. Additionally, the subjective perception of the observer can influence the findings. García-Resúa et al. showed that, although there was a significant correlation between classifications made by experienced observers based on Guillon’s schema, misinterpretations of the patterns might still occur, even within the same observer [14]. 

Despite the fact that Fodor et al. demonstrated that lower tear meniscus height measurements were more repeatable with Tearscope than slit-lamp biomicroscopy without staining in 31 healthy individuals, the subjectivity inherent in its use limits both its repeatability and its utility in comparison to other instruments that are more objective (Oculus^®^ Keratograph 5M and LipiView^®^) [37,38].

Finally, Tearscope presents a reliable and consistent method of increasing clinical observation and identification of ocular physiological alterations; yet, this automated procedure is susceptible to human error.

Cobra HD Camera (CSO, Florence, Italy) is a non-mydriatic digital fundus device with modules designed for retinal screening analysis. Additionally, it includes a dedicated module for meibography [17].

In a study conducted by Pult, the association between age, sex, and dry eye symptoms, as well as the quantification of meibomian gland (MG) loss, was investigated using a Cobra fundus camera. The study involved 112 participants and revealed substantial standard deviations in the mean MG loss between participants with and without dry eye symptoms (30 ± 17% and 45 ± 18%, respectively) [39].

Iphra and Gantz investigated the inter-session repeatability (ISR), inter-examiner reproducibility (IER), and within-subject variability (WSV) of the Cobra HD fundus camera meibographer. This study utilized Phoenix software. Participants were classified as either symptomatic or asymptomatic for dry eye based on their Ocular Surface Disease Index (OSDI) questionnaire scores. To determine the IER, seventy-four participants were evaluated on the same day by two examiners, referred to as Examiner 1 and Examiner 2. Subsequently, sixty-six of these participants were re-examined by Examiner 1 on a different date to calculate the ISR. The results showed that the Cobra HD fundus camera meibographer had good repeatability and reproducibility, and clinically similar findings should be obtained when used by different examiners on different occasions [40].

In conclusion, although the Cobra HD camera can only detect meibomian gland loss, it is useful for the meibographic assessment and follow-up of DED progression.

## 3. Discussion

Easy and rapid dry eye disease diagnosis is still a challenging unmet need in opthalmology. The algorithms proposed by various international societies and committees are frequently time-consuming and costly, and their use in the context of a busy medical setting is limited. Although 20% of our patients suffer from DED and ocular discomfort impact almost 40% of surgeons practice, a proper DED diagnostic method is still missed [41,42,43]. Therefore, several new diagnostic tools aim to fill this gap, making the diagnosis with a single “click”, although at a higher cost. 

The consistent use of these instruments in clinical settings may facilitate the diagnosis, tracking, and prevention of ocular surface diseases such as dry eye syndrome. However, the results in the literature are few and inconsistent. Most likely, there is no clear way for practitioners to use these automated tools to diagnose dry eyes in a standardized way. Furthermore, the lack of intra- and interobserver repeatability in certain measurement instruments limits neutrality and increases bias, influencing their use and distribution. A potential drawback of these imaging systems is their high cost, which can limit their accessibility in many healthcare centers.

## 4. Conclusions

No diagnostic instrument can replace the complex TFOS algorithm, and there is no single tool for a specific diagnosis, but research in this field is very active, and such primordial devices may be a promising reality in the very near future.

## Figures and Tables

**Figure 1 life-13-01425-f001:**
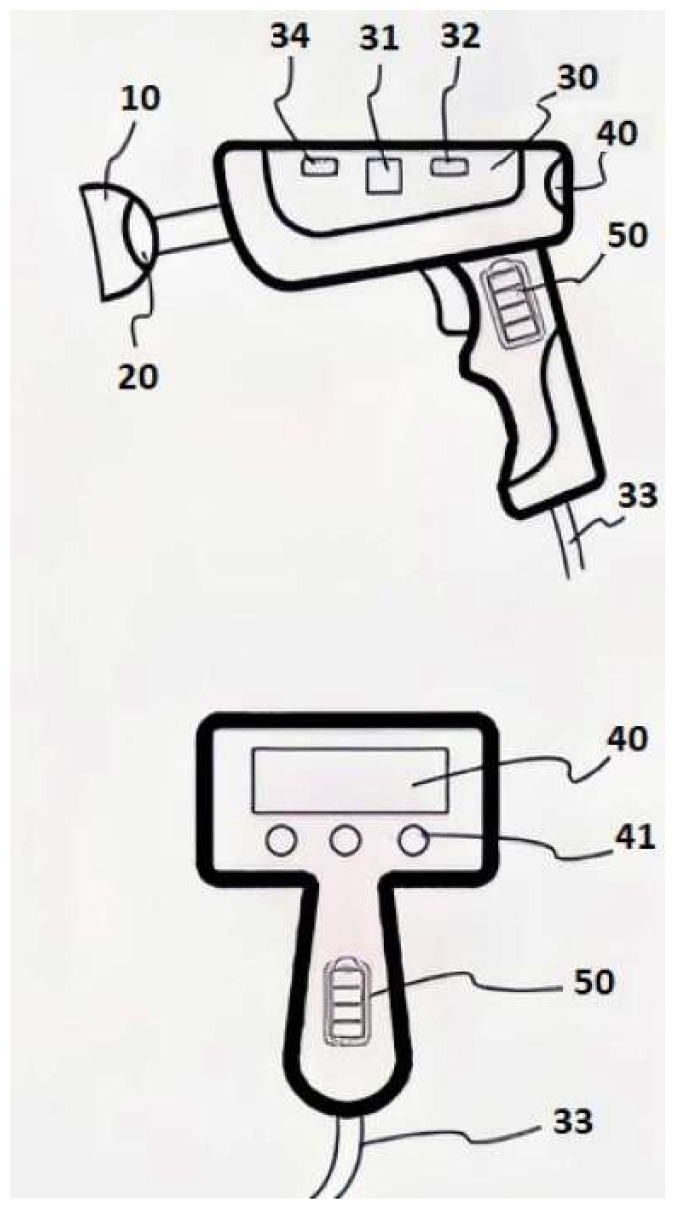
A schematic design of the diagnostic tool system. The figure shows a side and rear view of the schematic representation of a monosensor diagnostic prototype with: an eyepiece cup (10); the sensor (20) placed inside it; a processing board (30) equipped with processor (31), memory (34), and wireless connection device (32); the optional connection cable (33); a digital screen (40) with buttons (41); and a rechargeable power supply battery (50) placed in the handle [35].

**Table 1 life-13-01425-t001:** Ocular surface diagnostics devices comparison.

Device	Main Exams Performed											
	NIBUT	Osmolarity	LLT	TMH	Meibography	Eye Blink	Bulbar Redness	Inflammatory Evaluation	Demodex Presence	OSDI	Accurate Ocular Surface Images	RH
TearLab Osmolarity System^®^		x										
IDRA^®^	x		x	x	x	x						
Tearcheck^®^	x		x	x	x	x	x	x	x	x		
Keratograph 5M^®^	x		x	x	x		x					
I-PEN Osmolarity System^®^		x										
LipiView^®^II interferometer			x		x	x						
LacryDiag Ocular Surface Analyzer	x		x	x	x							
Cornea Dome Lens Imaging System^®^											x	
DEvice Hygrometer^©^												x
Tearscope-Plus^TM^	x		x	x								
Cobra HD Camera					x							

NIBUT, non-invasive tear break-up time; LLT, lipid layer thickness; TMH, tear meniscus height; RH, relative humidity.

## Data Availability

Data sharing not applicable.

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
