# Peer review of "Promising High-Tech Devices in Dry Eye Disease Diagnosis"

_life, 2023, doi:10.3390/life13071425_

Round 1

Reviewer 1 Report

This review outlines new instruments for diagnosis of dry eye disease based on a literature search. I think the authors would like their review to highlight the recommended instruments for busy clinical practices. However, they have failed to identify these. Therefore I would recommend a major revision of the manuscript with this outcome in mind. For example, the authors could improve the manuscript by including the literature that evaluates the repeatability and reproducibility of each instrument, if available. These outcomes are important when determining the utility of instruments used in comparative and observational studies, and clinical measurements used to diagnose diseases such as DED and MGD (Powell et al., 2012). Then, they could offer the reader a suggested instrument based on these findings. I believe this would make the manuscript very useful to clinicians.

The authors state in their conclusion “Furthermore, the lack of intra- and interobserver repeatability in some of the measurement instruments limits neutrality and increases bias, influencing their use and diffusion.”- which emphasizes even more so, the need for their inclusion of the repeatability and reproducibility for each instrument described in this review.

Powell DR, Nichols JJ, Nichols KK. Inter-examiner reliability in meibomian gland dysfunction assessment. Invest Ophthalmol Vis Sci. 2012;53:3120–5.

Major Comments:

I don’t agree with the authors’ statement regarding the application of triage question to detect signs of DED. The relationship between symptoms and signs of DED is not linear and varies across individuals and types of DED (Begley et al., 2003). The questions can assist in establishing the medical necessity for additional DED evaluation, and for monitoring the progression and response to treatments. In this regard, symptom measurements are very similar to clinical signs of DED. It is therefore recommended that a validated symptom questionnaire be administered at the beginning of the patient interaction.

 Begley C, Chalmers RL, Abetz L, Venkataraman K, Mertzanis P, Caffery BA, et al. The relationship between habitual patient-reported symptoms and clinical signs among patients with dry eye of varying severity. Invest Ophthalmol Vis Sci 2003;44:4753–4761.

I believe the authors meant to write DEQ-5 score ≥ 6 instead of DQ-5-6 and similarly OSDI score ≥ 13

The section referring to the diagnosis of DED should specifically state “Based on the TFOS DEWS II (2017) recommendations”

I think the authors should link between the paragraph stating the limitation of the TFOS DEWS II diagnosis algorithm within the clinical setting and the purpose of their review, which I believe was to examine if novel instruments could provide a diagnosis based on a set of less clinical tests that are incorporated in a single instrument.

“The tear-film contact took place at the temporal extent of the eyelid. The pen was docked into the reader after hearing the beep signaling successful tear collection [5].”- are the authors describing a study that they conducted? Why is this section worded in the past tense as if an experimental procedure is being described?

The various instruments detailed by the authors would be better placed under a subtitle that describes the measurement outcome variable. For example: the first two instruments measure “osmolarity” and could appear after a subtitle “Osmolarity”

Then, the definition and importance of the outcome variable should be described. In this case “Osmolarity is the entire concentration of dissolved particles in a solution, regardless of density, size, molecular weight, or electrical charges. Evaporation of tears, a reduction in tear production, and meibomian gland dysfunction all increase tear osmolarity, and tear osmolarity assessment is believed to be highly diagnostic for all types of DES [3]. Osmolarity is lower in the tear film of the exposed ocular surface area. Wind, cigarette smoke, indoor air conditioning and heat, as well as computer use when a subject does not blink enough, are all examples of conditions that can impair evaporation and thus tear osmolarity. Tear osmolarity was also found to correlate well with increased concentrations of inflammatory cytokines and matrix metalloproteinases (MMPs), as well as HLA-DR overexpression, suggesting that tear osmolarity may be useful in predicting ocular surface diseases associated with high levels of inflammatory mediators. [4].” Should go under the subtitle “Osmolarity” And then the instruments should be listed.

Nevertheless, Szalai et al. found significant overlap in tear osmolarity values measured with the TearLab system in the control and dry eye groups in their research, implying that this device should be used in association with traditional dry eye tests [8].- If there is overlap in values measured in normals vs. DED this does NOT indicate that the device should be used with other tests, it indicats that the instrument is not a reliable diagnostic test.

The authors state “Tear osmolarity measurement with the TearLab osmometer has high variability and fails to differentiate clinically diagnosed DED patients from healthy controls.” Followed by “As a result, TearLab is a quick and dependable tool in clinical practice”- dependable in what way? Its highly variable results?

I-PEN Osmolarity System® (I-MED Pharma Inc, Dollard-des-Ormeaux, Quebec, Canada) should be included here with the osmolarity instruments.

What do the authors mean by “IDRA must be placed between a slit lamp and a biomicroscope”- do the authors mean that it should be placed between the light source and the oculars? “Slit lamp biomicroscope” is typically used to describe the same instrument.

It enables fast and simple examinations, as a total of nine exams can be performed, providing breakthrough dry eye analysis in less than 10 minutes, report included. This sounds like it was extracted straight from an advertisement. Can the authors provide a reference for the time?

The OSDI should be described in the first instance of the manuscript and not within the instruments described. Further, grammatical suggestions are incorporated below:

“The latter is a validated questionnaire that is frequently used in clinical trials because it allows for a quick evaluation of dry eye disease (DED) and its effect on patient quality of life (QoL) [15]. The OSDI includes 12 questions designed to assess patients' symptoms in the previous week. It is divided into three sections: frequency of symptoms, effect on vision-related QoL, and presence of any environmental triggers [15]. Participants select one of five options for each topic, each scored on a range of 0 to 4: None of the time (0), some of the time (1), half of the time (2), the majority of the time (3), and the entire time (4). The final number is determined on a scale of 0 to 100, with higher scores indicating greater disability. A score between 0 and 12 is considered normal, while a score between 13 and 22 indicates mild disease, a score between 23 and 32 indicates moderate DED, and any number between 33 and 100 indicates severe DED [15].”

Further, the OSDI does not discriminate between evaporative and aqueous-deficiency (Machalinska et al., 2016) . The authors should state this.

A. Machalińska, A. Zakrzewska, K. Safranow, B. Wiszniewska, and B. Machaliński, ‘Risk Factors and Symptoms of Meibomian Gland Loss in a Healthy Population’, J Ophthalmol, vol. 2016, pp. 1–8, 2016, doi: 10.1155/2016/7526120.

The authors should add that the keratograph has substantial examiner bias (Garduno et al., 2021) on the one hand, but has been reported to have good inter-examiner reproducibility (mean difference between examiners of 0.08 ± 0.55 and 0.13 ± 0.50 grade units in two separate sessions, respectively) with low within-subject variability (95% limits of agreement for two different examiners of −1.02 to +1.10 and −1.27 to +1.09 grade units, respectively- Ngo et al., 2014) on the other hand.

Garduño F, Salinas A, Contreras K, Rios Y, García N, Quintanilla P, et al. Comparative study of two infrared meibographers in evaporative dry eye versus nondry eye patients. Eye Contact Lens. 2021;47:335–40.

Ngo W, Srinivasan S, Schulze M, Jones L. Repeatability of grading meibomian gland dropout using two infrared systems. Optom Vis Sci. 2014;91:658–67.

Why did the authors only include the keratograph? There are other options, perhaps newer ? Wasn’t the point of the review to detect more recent instruments? For example, why wasn’t the Sirius Scheimpflug included?

Regarding the findings among people with a high BMI where osmolarity was reported to be significantly associated between two method, the authors applied a Bland and Altman analysis. In a review, the authors are to critically review the reported study outcomes. Therefore the authors should state the interchangeability in terms of the clinical expected outcomes and limits of agreement.

In their conclusions, I think the authors intended to write that an instrument that incorporates the clinical outcome measures listed in the TFOS DEWS II report is warranted.

Minor Comments:

Throughout the document: the authors should spell out the first instance of an abbreviation (DEQ, OSDI, HLA-DR)

Introduction

Please add signs indicating the direction and amount considered abnormal as stated for other clinical signs: (NIBUT 10 sec), elevated tear osmolarity (308 mOsm/L) or significant inter-eye disparity in osmolarity (8 mOsm)

Methods

What is the range of years for the literature search?

Did the authors translate papers that were not written in the English language?

Provide a reference for this sentence: Osmolarity is lower in the tear film of the exposed ocular surface area. Wind, cigarette smoke, indoor air conditioning and heat, as well as computer use when a subject does not blink enough, are all examples of conditions that can impair evaporation and thus tear osmolarity.

The OcuSense TearLabTM osmometer (OcuSense Inc., San Diego, CA, USA)- should be bold and have
“:” after the description similarly to the instrument above it.

Dorota H. Szczesna-Iskander- I think just the last name is necessary

IDRA- Add IDRA ocular surface analyzer

Add IDRA here and replace research with study: A prospective study found good concordance between the IDRA ocular surface analyzer and routine diagnostic procedure measurements in distinguishing between normal and DED subjects [11].

The grades are converted to nanometers

What is the meaning of this sentence: DRA has a maximum cutoff wavelength of 100 nm [14].

however, the small sample size affects the few findings obtained.- which small sample size? Of which study?

I have not heard of something being strongly weak. Perhaps “very weak”?

Dong Hui Lim et al. Conclude- Please state last name only. Change to “concluded”

However, additional studies with larger sample sizes are necessary- how many participants were included in the studies detailed by the authors?

The algorithms proposed by various international societies and committees- are there others aside from TFOS?

 are frequently time-consuming and costly, and they are rarely applicable in busy daily clinical practice.- provide a reference for this claim

the results in the literature are few and inconsistent- What do the authors mean? Unclear

repeatability in some of the measurement instruments limits neutrality and increases bias, influencing their use and diffusion- what do the authors mean by diffusion ? Distribution?

All comments listed above

Author Response

We have given careful consideration to comments of the reviewer and have revised the manuscript to address those concerns. We would like to heartily thank the reviewer for the help given in improving our manuscript and better presenting our review.

I don’t agree with the authors’ statement regarding the application of triage question to detect signs of DED. The relationship between symptoms and signs of DED is not linear and varies across individuals and types of DED (Begley et al., 2003). The questions can assist in establishing the medical necessity for additional DED evaluation, and for monitoring the progression and response to treatments. In this regard, symptom measurements are very similar to clinical signs of DED. It is therefore recommended that a validated symptom questionnaire be administered at the beginning of the patient interaction.

We thank the reviewer for the suggestion, and we modified the text: “The first phase in the process of diagnosing dry eye disease involves the utilization of triage questions which could establish the need for additional DED evaluation, and could exclude disorders such as conjunctivitis, blepharitis, Sjögren syndrome, infection, and lid-related disease.”

I believe the authors meant to write DEQ-5 score ≥ 6 instead of DQ-5-6 and similarly OSDI score ≥ 13.

We thank the reviewer; we correct the mistakes.

The section referring to the diagnosis of DED should specifically state “Based on the TFOS DEWS (2017) recommendations.”

We thank reviewer for the suggestion, and we added the sentence.

I think the authors should link between the paragraph stating the limitation of the TFOS DEWS II diagnosis algorithm within the clinical setting and the purpose of their review, which I believe was to examine if novel instruments could provide a diagnosis based on a set of less clinical tests that are incorporated in a single instrument.

We thank reviewer for the advice. We improved the structure of the paragraph.

“The tear-film contact took place at the temporal extent of the eyelid. The pen was docked into the reader after hearing the beep signaling successful tear collection [5].”- are the authors describing a study that they conducted? Why is this section worded in the past tense as if an experimental procedure is being described?

We thank the reviewer for the suggestion. We changed the verb tense in the text.

The various instruments detailed by the authors would be better placed under a subtitle that describes the measurement outcome variable. For example: the first two instruments measure “osmolarity” and could appear after a subtitle “Osmolarity”

Then, the definition and importance of the outcome variable should be described. In this case “Osmolarity is the entire concentration of dissolved particles in a solution, regardless of density, size, molecular weight, or electrical charges. Evaporation of tears, a reduction in tear production, and meibomian gland dysfunction all increase tear osmolarity, and tear osmolarity assessment is believed to be highly diagnostic for all types of DES [3]. Osmolarity is lower in the tear film of the exposed ocular surface area. Wind, cigarette smoke, indoor air conditioning and heat, as well as computer use when a subject does not blink enough, are all examples of conditions that can impair evaporation and thus tear osmolarity. Tear osmolarity was also found to correlate well with increased concentrations of inflammatory cytokines and matrix metalloproteinases (MMPs), as well as HLA-DR overexpression, suggesting that tear osmolarity may be useful in predicting ocular surface diseases associated with high levels of inflammatory mediators. [4].” Should go under the subtitle “Osmolarity” And then the instruments should be listed.

We thank the reviewer for the advice, but many instruments analyze different variables, making it difficult to put them into a specific subtitle.

Nevertheless, Szalai et al. found significant overlap in tear osmolarity values measured with the TearLab system in the control and dry eye groups in their research, implying that this device should be used in association with traditional dry eye tests [8].- If there is overlap in values measured in normals vs. DED this does NOT indicate that the device should be used with other tests, it indicats that the instrument is not a reliable diagnostic test.

We thank the reviewer for the suggestion. We modified the text: “Nevertheless, Szalai et al. found significant overlap in tear osmolarity values measured with the TearLab system in the control and dry eye groups in their research, implying that measurement of tear osmolarity utilizing the TearLab osmometer is highly variable and does not distinguish individuals diagnosed with dry eye disease from healthy controls.”

The authors state “Tear osmolarity measurement with the TearLab osmometer has high variability and fails to differentiate clinically diagnosed DED patients from healthy controls.” Followed by “As a result, TearLab is a quick and dependable tool in clinical practice”- dependable in what way? Its highly variable results?

We thank reviewer for highlighting it. We clarified it in the text.

What do the authors mean by “IDRA must be placed between a slit lamp and a biomicroscope”- do the authors mean that it should be placed between the light source and the oculars? “Slit lamp biomicroscope” is typically used to describe the same instrument.

We thank reviewer for highlighting it. We clarified it in the text.

The OSDI should be described in the first instance of the manuscript and not within the instruments described. Further, grammatical suggestions are incorporated below:

“The latter is a validated questionnaire that is frequently used in clinical trials because it allows for a quick evaluation of dry eye disease (DED) and its effect on patient quality of life (QoL) [15]. The OSDI includes 12 questions designed to assess patients' symptoms in the previous week. It is divided into three sections: frequency of symptoms, effect on vision-related QoL, and presence of any environmental triggers [15]. Participants select one of five options for each topic, each scored on a range of 0 to 4: None of the time (0), some of the time (1), half of the time (2), the majority of the time (3), and the entire time (4). The final number is determined on a scale of 0 to 100, with higher scores indicating greater disability. A score between 0 and 12 is considered normal, while a score between 13 and 22 indicates mild disease, a score between 23 and 32 indicates moderate DED, and any number between 33 and 100 indicates severe DED [15].”

We thank the reviewer for the suggestion. We reorganized the text.

Further, the OSDI does not discriminate between evaporative and aqueous-deficiency (Machalinska et al., 2016). The authors should state this.

We thank reviewer for the advice. It was mentioned in the text.

It enables fast and simple examinations, as a total of nine exams can be performed, providing breakthrough dry eye analysis in less than 10 minutes, report included. This sounds like it was extracted straight from an advertisement. Can the authors provide a reference for the time?

We agree with the reviewer, and we improved the structure of the sentence.

The authors should add that the keratograph has substantial examiner bias (Garduno et al., 2021) on the one hand, but has been reported to have good inter-examiner reproducibility (mean difference between examiners of 0.08 ± 0.55 and 0.13 ± 0.50 grade units in two separate sessions, respectively) with low within-subject variability (95% limits of agreement for two different examiners of −1.02 to +1.10 and −1.27 to +1.09 grade units, respectively- Ngo et al., 2014) on the other hand.

We thank the reviewer for the suggestion. We added it in the text.

Throughout the document: the authors should spell out the first instance of an abbreviation (DEQ, OSDI, HLA-DR)

We agree with the reviewer, and we spelt out the first instance of the abbreviations throughout the text.

What is the range of years for the literature search?

We thank reviewer for highlighting it. We clarified it in the text.

Did the authors translate papers that were not written in the English language?

It is stated in the methods that the search was conducted without without any restriction of language.

Provide a reference for this sentence: Osmolarity is lower in the tear film of the exposed ocular surface area. Wind, cigarette smoke, indoor air conditioning and heat, as well as computer use when a subject does not blink enough, are all examples of conditions that can impair evaporation and thus tear osmolarity.

We agree with the reviewer, and we added the reference.

Dorota H. Szczesna-Iskander- I think just the last name is necessary.

We appreciate the suggestion; we modified it.

IDRA- Add IDRA ocular surface analyzer.

We agree with the reviewer, and we added it.

Add IDRA here and replace research with study: A prospective study found good concordance between the IDRA ocular surface analyzer and routine diagnostic procedure measurements in distinguishing between normal and DED subjects [11].

We appreciate the suggestion; we modified the sentence.

The grades are converted to nanometers.

We apologize for the mistake, and we corrected it.

However, the small sample size affects the few findings obtained - which small sample size? Of which study?

We thank the reviewer for the suggestion. We modified the sentence.

I have not heard of something being strongly weak. Perhaps “very weak”?

We thank the reviewer for the suggestion. We modified the sentence.

Dong Hui Lim et al. Conclude- Please state last name only. Change to “concluded”

We apologize for the mistake, and we corrected it.

The algorithms proposed by various international societies and committees- are there others aside from TFOS?

We thank reviewer for highlighting it.  There are additional algorithms for the diagnosis and treatment of DED aside from TFOS, such as CEDARS Algorithm and ASCRS Algorithm.

Repeatability in some of the measurement instruments limits neutrality and increases bias, influencing their use and diffusion- what do the authors mean by diffusion? Distribution?

We thank the reviewer for the suggestion. We changed “diffusion” with “distribution”.

Reviewer 2 Report

The author reviewed papers regarding DED, which supplied potential screenning solutions in detecting DED. Here are my commonts:

1. Abbreviations should be written in full the first time they appear. For example: TFOS DEWS.  

2. In inthroduction section, the author stated that TFOS DEWS owns a complete test and contains a number of steps, so they performed current study. When the author reviewed several DED detecting methods, most of them contains less steps than that of TFOS DEWS. It is acceptable that less steps but time saving, instead of just make a conclusion that TFOS was unreplacable. So the author should focus on when these methods were mainly chosed in what situation.

3. All the literature reviews were descriptive without any statistical analysis or data to support them.  If the authors waant to proof that "TFOS was unreplacable", the current study should provide data to support this. The current version seems somewhat subjective.

4. The current paper does not include a results section and a discussion section.

5. Tables should be self-explanatory. It's hard to understand what Table 1 is trying to tell the reader. 

Author Response

We have given careful consideration to comments of the reviewer and have revised the manuscript

to address those concerns. We would like to heartily thank the reviewer for the help given in improving our manuscript and better presenting our review.

Abbreviations should be written in full the first time they appear. For example: TFOS DEWS

We agree with the reviewer, and we written in full all the abbreviations the first time they appear throughout the text.

 In inthroduction section, the author stated that TFOS DEWS owns a complete test and contains a number of steps, so they performed current study. When the author reviewed several DED detecting methods, most of them contains less steps than that of TFOS DEWS. It is acceptable that less steps but time saving, instead of just make a conclusion that TFOS was unreplacable. So the author should focus on when these methods were mainly chosed in what situation.

All the literature reviews were descriptive without any statistical analysis or data to support them.  If the authors waant to proof that "TFOS was unreplacable", the current study should provide data to support this. The current version seems somewhat subjective.

We thank reviewer for highlighting it. We don’t think that TFOS algorithm is “unreplacable”, but it is time-consuming and costly, and its use in the context of a busy medical setting is limited. Hence, the use of new instruments in clinical settings may facilitate the diagnosis of ocular surface diseases like dry eye syndrome. However, we conclude that no diagnostic instrument can replace the complex TFOS algorithm, and there is no single tool for a specific diagnosis.

The current paper does not include a results section and a discussion section.

We thank the reviewer for the suggestion. We added a discussion section in the text.

Tables should be self-explanatory. It's hard to understand what Table 1 is trying to tell the reader. 

We thank the reviewer for the advice, we better organized Table 1.

Round 2

Reviewer 1 Report

Major comments:

The authors have addressed some of the previous comments and suggestions.

Unfortunately, the authors did not include the Sirium Scheimflug or the Cobra HD fundus camera meibographers in their manuscript as previously requested.

In the introduction the concluding sentence states “These systems claim to have several benefits over traditional methods of diagnosis, such as being non-invasive, providing standardized and objective results, ena-bling the monitoring of disease progression and treatment effectiveness, being user-friendly, and enabling rapid task execution” – does the review cover the benefits? Also the authors were asked to include a description about the reliability of the methods presented. This should probably also be stated in the introduction.

In many instances, the authors state that more studies are needed with larger sample sizes, however they do not detail the number of participants in the studies that they describe.

The authors describe results of studies too generally. More information should be provided in terms of the number of participants, the outcome measure that was compared, and the mean differences. Just stating that there were significant differences in a review paper is not sufficient. The authors should state how they were different? Which instrument measured lower or higher values? And by how much. This allows the reader to make informed decisions about the findings presented.

Minor comments:

Page 1-

 TFOS DEWS II (Tear Film and Ocular Surface Society TFOS Interna-tional Dry Eye Workshop II, 2017) should appear in the first instance of TFO DEWS II and not the second one

Add the appropriate sign: such as decreased tear film stability (NIBUT, Non Invasive Tear Break Up-Time< 10 sec), elevated tear osmolarity (>308 mOsm/L) or significant inter-eye disparity in osmolarity (>8 mOsm/L),

 Sentences missing a reference: 

 The incidence of DED exhibits a positive correlation with advancing age and varies between five percent and fifty percent across the overall population.

 DED is characterized by a range of symptoms such as ocular pain, burning, stinging, discomfort, a foreign body sensation, poor visual acuity, photophobia, and irritation. The

Page 2:

It is organized

Participants have to select one
or even better: Participants are asked to select

Each participant underwent evaluation on a scale ranging from 0 to 4. Better phrased :” Each participant is evaluated using a scale ranging from 0 to 4” though I am unclear and think the authors mean that each item on the questionnaire is scored between 0 and 4.

overall numerical value is calculated

It is possible that the TFOS DEWS II (Tear Film and Ocular Surface Society TFOS In-ternational Dry Eye Workshop II) diagnostic-  no need to spell everything out here as it was previously explained

Page 4

A prospective study with how many participants?

The device facilitates quick evaluations, that includes nine examinations: non-invasive break time, tear film stability, ocular surface inflammatory assessment, meibography IR, Demodex, eye redness, abortive blinking, tear meniscus height, and the OSDI questionnaire.  This results in a simple dry eye analysis.

Page 5

has significant examiner bias- are there any numbers to support this? What is the reference for this

but on the other hand, it has been reported to have strong inter-examiner reproducibility with low within-subject variability – are there numbers to support this?

Page 6

Furthermore, they showed that the I-PEN system was less effective than the TearLab Osmolarity System [7] [27] [28].- less effective in what way?

a highly trained researcher or clinician?

results were much higher- how much higher exactly?

it is limited in its ability to assess LLT values that are greater than 100 nm.- missing a reference

In conclusion, the data obtained from the LVII LLT should be compared to other instruments. Instead of measurements

However, additional studies with larger sample sizes are necessary.- how many were included in the studies described by the authors?

saving photos for later use- should be modified to “save photos for later use”

who are “these different instruments and who are “other innovative devices”- please elaborate

Their findings revealed a strong relationship between the devices, but low agreement for any individual observer.  A strong relationship between which measure? Low agreement for which measure? How low? Can the authors provide more details?

Therefore, the authors concluded instead of “Hence, the authors propose”

Page 7:

Eye movements instead of ocular movements

Moreover should be replaced with “However”

DEvice Hygrometer, the “E” is capitol on purpose?

Following therapy is guided by the utilization of simple algorithms.- grammar incorrect, I don’t understand.

The implementation of a non-contact approach eliminates the potential bias that may arise from the closeness of conventional testing methods to the ocular surface. Replace “closeness” with “proximity”. Provide a reference for the claim that proximity between an instrument and the ocular surface produces bias,

and safe (no contact) instrument for measurement of tear films- change to and safer (as it is non-contact) instrument for measurement of tear film.

In the list of minor suggestions I listed possible edits that will improve the readability of the paper. However, there were at least two instances of sentences that were incomprehensible to me and I could not offer suggested revisions and a professional editor should be consulted.

Author Response

We have given careful consideration to comments of the reviewer and have revised the manuscript to address those concerns. We would like to heartily thank the reviewer for the help given in improving our manuscript.

Major comments:

The authors have addressed some of the previous comments and suggestions.

Unfortunately, the authors did not include the Sirium Scheimflug or the Cobra HD fundus camera meibographers in their manuscript as previously requested.

We thank reviewer for highlighting it. We included Cobra HD fundus camera meibographer in our manuscript.

In the introduction the concluding sentence states “These systems claim to have several benefits over traditional methods of diagnosis, such as being non-invasive, providing standardized and objective results, ena-bling the monitoring of disease progression and treatment effectiveness, being user-friendly, and enabling rapid task execution” – does the review cover the benefits? Also the authors were asked to include a description about the reliability of the methods presented. This should probably also be stated in the introduction.

We appreciate the suggestion. We included some information in the introduction and offered more data about the device's reliability through a more detailed discussion of the studies cited in the text.

In many instances, the authors state that more studies are needed with larger sample sizes, however they do not detail the number of participants in the studies that they describe.

We thank the reviewer for the suggestion, and we provided the number of partecipants in most of the studies.

The authors describe results of studies too generally. More information should be provided in terms of the number of participants, the outcome measure that was compared, and the mean differences. Just stating that there were significant differences in a review paper is not sufficient. The authors should state how they were different? Which instrument measured lower or higher values? And by how much. This allows the reader to make informed decisions about the findings presented. 

We thank reviewer for the advice. We provided more information about the studies described in the manuscript.

Minor comments:

Page 1-

 TFOS DEWS II (Tear Film and Ocular Surface Society TFOS Interna-tional Dry Eye Workshop II, 2017) should appear in the first instance of TFO DEWS II and not the second one

We thank the reviewer for the suggestion, and we modified it in the text.

Add the appropriate sign: such as decreased tear film stability (NIBUT, Non Invasive Tear Break Up-Time< 10 sec), elevated tear osmolarity (>308 mOsm/L) or significant inter-eye disparity in osmolarity (>8 mOsm/L)

We thank reviewer for the advice. We add it in the text.

 Sentences missing a reference:  

 The incidence of DED exhibits a positive correlation with advancing age and varies between five percent and fifty percent across the overall population.

We thank reviewer for the advice. We added the reference.

 DED is characterized by a range of symptoms such as ocular pain, burning, stinging, discomfort, a foreign body sensation, poor visual acuity, photophobia, and irritation. The

We thank reviewer for the advice. We added the reference.

Page 2: 

It is organized

We appreciate the suggestion; we modified it.

Participants have to select one
or even better: Participants are asked to select

Each participant underwent evaluation on a scale ranging from 0 to 4. Better phrased :” Each participant is evaluated using a scale ranging from 0 to 4” though I am unclear and think the authors mean that each item on the questionnaire is scored between 0 and 4.

We thank the reviewer for the suggestion. We reorganized the text.

overall numerical value is calculated

We appreciate the suggestion; we modified it.

It is possible that the TFOS DEWS II (Tear Film and Ocular Surface Society TFOS In-ternational Dry Eye Workshop II) diagnostic-  no need to spell everything out here as it was previously explained

We thank reviewer for highlighting it. We deleted it.

Page 4

A prospective study with how many participants?

We thank reviewer for the advice. We add it in the text.

The device facilitates quick evaluations, that includes nine examinations: non-invasive break time, tear film stability, ocular surface inflammatory assessment, meibography IR, Demodex, eye redness, abortive blinking, tear meniscus height, and the OSDI questionnaire.  This results in a simple dry eye analysis.

We thank the reviewer for the suggestion. We improved the structure of the sentences.

Page 5

has significant examiner bias- are there any numbers to support this? What is the reference for this

but on the other hand, it has been reported to have strong inter-examiner reproducibility with low within-subject variability – are there numbers to support this?

We thank the reviewer for the suggestion. We added the reference and provided more details.

Page 6

Furthermore, they showed that the I-PEN system was less effective than the TearLab Osmolarity System [7] [27] [28].- less effective in what way?

We thank reviewer for the advice. We provided more details in the text.

a highly trained researcher or clinician?

We thank reviewer for the advice. We modified it.

results were much higher- how much higher exactly?

We thank the reviewer for the suggestion. We clarified it in the text.

it is limited in its ability to assess LLT values that are greater than 100 nm.- missing a reference

We thank reviewer for highlighting it. We added the reference.

In conclusion, the data obtained from the LVII LLT should be compared to other instruments. Instead of measurements

We thank reviewer for highlighting it. We modified it.

However, additional studies with larger sample sizes are necessary.- how many were included in the studies described by the authors?

We thank the reviewer for the suggestion. We clarified it in the text.

saving photos for later use- should be modified to “save photos for later use”

We thank reviewer for highlighting it. We modified it.

who are “these different instruments” and who are “other innovative devices”- please elaborate

We thank the reviewer for the suggestion. We clarified it in the text.

Their findings revealed a strong relationship between the devices, but low agreement for any individual observer.  A strong relationship between which measure? Low agreement for which measure? How low? Can the authors provide more details?

We thank reviewer for highlighting it. We provided more details in the text

Therefore, the authors concluded instead of “Hence, the authors propose”

We thank reviewer for highlighting it. We modified it.

Page 7: 

Eye movements instead of ocular movements

We thank the reviewer for the suggestion. We modified it.

Moreover should be replaced with “However”

We thank the reviewer for the suggestion. We replaced it.

DEvice Hygrometer, the “E” is capitol on purpose?

We thank reviewer for the advice. DEvice Hygrometer is the device name.

Following therapy is guided by the utilization of simple algorithms.- grammar incorrect, I don’t understand.

We thank the reviewer for the suggestion. We modified the sentence.

The implementation of a non-contact approach eliminates the potential bias that may arise from the closeness of conventional testing methods to the ocular surface. Replace “closeness” with “proximity”. Provide a reference for the claim that proximity between an instrument and the ocular surface produces bias,

and safe (no contact) instrument for measurement of tear films- change to and safer (as it is non-contact) instrument for measurement of tear film.

We thank the reviewer for the suggestion. We modified the sentences.

Comments on the Quality of English Language

In the list of minor suggestions I listed possible edits that will improve the readability of the paper. However, there were at least two instances of sentences that were incomprehensible to me and I could not offer suggested revisions and a professional editor should be consulted.

We thank the reviewer for the suggestion. Manuscript was furtherly revised by a native English speaker.

Reviewer 2 Report

Thank you for your point-to-point responses. I have no further comments regarding this paper.

Author Response

We would like to thank the reviewer for the help given in improving our manuscript